# Impact of COVID-19 pandemic related to lockdown measures on tropospheric NO$_2$ columns over Île-de-France

Andrea Pazmiño[1], Matthias Beekmann[2], Florence Goutail[1], Dmitry Ionov[3], Ariane Bazureau[1], Manuel Nunes-Pinharanda[1], Alain Hauchecorne[1], Sophie Godin-Beekmann[1]

[1]LATMOS/IPSL, UVSQ, Université Paris-Saclay, Sorbonne Université, CNRS, Guyancourt, France
[2]LISA/IPSL, UMR CNRS 7583, Université Paris Est Créteil, Université de Paris, Créteil, France
[3]University of Saint Petersburg, Saint Petersburg, Russia

*Correspondence to*: Andrea Pazmiño (andrea.pazmino@latmos.ipsl.fr)

**Abstract.** The evolution of NO$_2$, considered as proxy for air pollution, was analyzed to evaluate the impact of 1$^{st}$ lockdown (March 17$^{th}$ – May 10$^{th}$ 2020) over île-de-France region (Paris and surroundings). Tropospheric NO$_2$ columns measured by two UV-Visible SAOZ spectrometers were analyzed to compare the evolution of NO$_2$ between urban and suburban sites during the lockdown. The urban site is the observation platform QUALAIR (48°50'N/2°21'E) on the Pierre et Marie Curie Campus of Sorbonne University in the center of Paris. The suburban site is located at Guyancourt (48°46'N/2°03'E), University of Versailles Saint Quentin, 24 km south-west of Paris. Tropospheric NO$_2$ columns above Paris and Guyancourt have shown similar values during the whole lockdown period from March to May 2020. One decade datasets were filtered to consider air masses at both sites with similar meteorological conditions. The median NO$_2$ columns, as well as the surface measurements of AIPARIF (Air Quality Observatory in Ile de France) during the lockdown period in 2020 were compared to the extrapolated values estimated from a linear trend analysis for the 2011-2019 period at each station. Negative NO$_2$ trends of -1.5 Pmolec cm$^{-2}$ yr$^{-1}$ (~-6.3 % yr$^{-1}$) are observed from the columns and of -2.2 µg m$^{-3}$ yr$^{-1}$ (~-3.6 % yr$^{-1}$) from the surface concentration.

The negative anomaly in tropospheric columns in 2020 attributed to lockdown (and related emission reductions) was found to be 56% at Paris and 46% at Guyancourt, respectively. Similar anomaly was found in the data of surface concentrations, amounting for 53% and 28% at the urban and suburban sites, accordingly.

## 1 Introduction

Megacities can be considered as a hot spot of anthropogenic pollution due to the concentration of population and human activities. People living in urban areas are exposed to air quality levels that often exceed the World Health Organization (WHO) recommended limits (WHO, 2006). In 2020, the emergence of a novel coronavirus that causes the COVID-19 disease in many countries around the world has prompted the governments of the affected states to apply restrictive regulations. Most countries implemented lockdown measures (restrictions on people movements) to limit the progression of

the COVID-19 pandemic. As a result, urban areas have become interesting "laboratories" for analyzing the impact of these measures on air quality. Atmospheric concentrations of air pollutants in megacities were expected to decrease as a direct impact of air and road traffic activity drop during the lockdown period. Observations of TROPOMI instrument onboard the Copernicus Sentinel 5-Precursor (S5P) satellite (Veefkind et al., 2012) were the earliest ones to be presented by the media to

show the significant decrease of tropospheric $NO_2$ columns in the Hubei province in China (20-50% in urban areas, Ding et al., 2020), the first region affected by the COVID-19 in December 2019. Indeed, tropospheric $NO_2$ is considered as a good proxy for $NO_x$ ($NO_x=NO+NO_2$) concentrations since NO is rapidly converted into $NO_2$ by the photochemical cycle involving tropospheric ozone. NOx levels are directly linked to human activities, for example over the Ile-de-France region, in which the Greater Paris region is imbedded, and for the year 2018, road traffic contributes to 53% of NOx emissions,

followed by industry (13%, including also energy and waste treatment), residential heating (11%) and airports (9%) (https://www.airparif.asso.fr/surveiller-la-pollution/les-emissions, last consulted in August 2021).

Many studies have focused on $NO_2$ reductions due to lockdowns in 2020 at specific cities in China (Ding et al., 2020, Griffith et al., 2020), and in other affected countries (Bauwens et al., 2020, Prunet et al., 2020) using only satellite observations (Bauwens et al., 2020, Koukouli et al., 2020, Liu et al., 2020) or additionally ground-based instruments (Biswal

et al., 2020, Prunet et al., 2020). Other studies analyzed the lockdown period using in situ monitoring networks in the cities (Baldasano, 2020, Biswal et al., 2020, Krecl et al., 2020). Model simulations were also analyzed to assess the respective $NO_2$ decreases (Koukouli et al., 2020, Liu et al., 2020, Menut et al., 2020).

The objective of this study is to quantify the effect of $NO_2$ decreases due to lockdown considering long-term variability and meteorological conditions over Ile-de-France region during the last decade using different datasets characterizing the

lockdown impact at local scale with in situ instrumentation, and at larger scale including a large part of the agglomeration with tropospheric column measurements. Two complementary sites are used, one in the center of Paris and the other one in the peripheral zone to highlight the possibly heterogeneous impact of lockdown in Ile de France region. The originality of the study is to rely not only on a single reference year before the COVID-19 pandemic that could strongly bias the study, but on a long decadal data set, in order to account for $NO_2$ variability on a longer period. This allows in addition calculating long

term $NO_2$ column changes over the Paris region. Specific data filtering using wind speed and direction is applied in order to isolate data, which are affected by local pollution in the Greater Paris area, and to consider the changes in meteorological conditions for the different years.

This paper is organized as follows. Observations of tropospheric and surface amounts of $NO_2$ by ground-based and satellite measurements are presented in Sect. 2 as well as the wind data from European Reanalysis. The description of the method

used to discriminate specific data to calculate $NO_2$ decrease in 2020 taking into account similar meteorological conditions is presented in Sect. 3. The results of $NO_2$ decreases in 2020 due to lockdown are shown in Sect. 4 for the different datasets. The results of $NO_2$ level reductions in respect to literature findings are discussed in Sect. 5. Conclusions are finally presented in Sect. 6.

**2 NO₂ Data**

Tropospheric NO$_2$ columns measured by two ground-based SAOZ instruments were analyzed to trace and intercompare the evolution of NO$_2$ in the urban and suburban regions of Ile-de-France. The analysis was supplemented by a study of NO$_2$ column satellite measurements using the TROPOMI instrument. In addition, the in-situ measurements of NO$_2$ surface concentrations from the AIRPARIF air quality network were also considered. In this work, the ten years period 2011-2020, with the first year corresponding to the start of SAOZ measurements at the suburban site of Guyancourt was considered.

Table 1 shows the ground-based stations, type of instrument and geographical coordinates and Figure 1 the location of each station in Ile-de-France region.

**Table 1: Ground-based stations used in this study: station, place, instrument and geographical coordinates.**

| Station | Place | Instrument | Lat/Lon |
|---------|-------|------------|---------|
| Paris | QUALAIR, Sorbonne-Université, Paris 5th | SAOZ | 48°50'N/2°21'E |
| Guyancourt | LATMOS, Guyancourt | SAOZ | 48°46'N/2°03'E |
| CELES | Quai des Célestins, Paris 5th | AIRPARIF | 48°51'N/2°21'E |
| PA13 | Parc de Choisy, park in Paris 13th | AIRPARIF | 48°49'N/2°21'E |
| PA07 | Allée des Refuzniks, Paris 7th | AIRPARIF | 48°51'N/2°17'E |
| EIFF | 300 m top of Eiffel Tower, Paris 7th | AIRPARIF | 48°51'N/2°17'E |
| VERS | Versailles | AIRPARIF | 48°48'N/2°08'E |

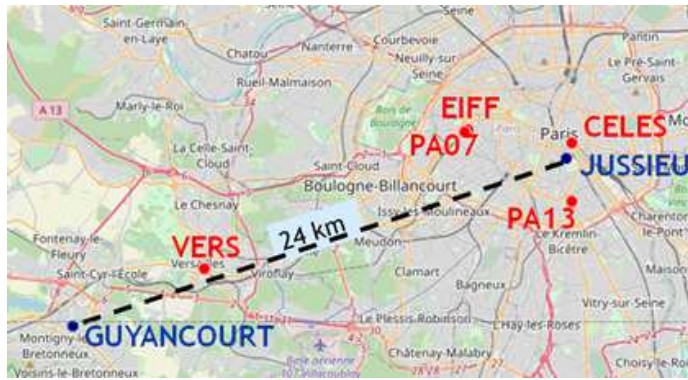

**Figure 1: Locations of the AIRPARIF (red points) and SAOZ (blue points) stations. Black dash line corresponds to the distance between both SAOZ stations. Map data © OpenStreetMap contributors 2021. Distributed under the Open Data Commons Open Database License (ODbL) v1.0.**

### 2.1 Tropospheric columns

#### 2.1.1 SAOZ data

The $NO_2$ tropospheric columns at Ile-de-France region are measured by two ground-based SAOZ (Système d'Analyse par Observation Zénithale) instruments (Pommereau and Goutail, 1988) that are part of French research infrastructure ACTRIS. The first one was installed in 2005 at the observation platform QUALAIR (http://qualair.aero.jussieu.fr/) of Sorbonne University in Paris (urban station) and the second one is operational at LATMOS laboratory in Guyancourt (South-West suburban station) since 2011. SAOZ is a UV-Visible spectrometer primary designed for monitoring stratospheric ozone and $NO_2$ during twilight observations in the frame of NDACC (Network for the Detection of Atmospheric Composition Change) (see Hendrick et al., 2011 for a description of retrieval). The long-term data series of SAOZ instruments were compared with data from most satellite missions to validate or monitor their performance. For example, SAOZ instruments participated in the validation of the latest satellite mission Sentinel 5 Precursor launched on October 2017 for the measurements of ozone (Garane et al., 2019) and stratospheric $NO_2$ (Verhoelst et al., 2021) columns.

During the day, SAOZ observations are sensitive to increased tropospheric $NO_2$ amounts in polluted regions (Tack et al., 2015). Every ~2 minutes, the sunlight backscattered by the atmosphere in the zenith direction of SAOZ is acquired and the DOAS (Differential Optical Absorption Spectroscopy) method (Platt and Stutz, 2008) is applied in the $NO_2$ absorptions bands to obtain the respective slant column densities. The stratospheric $NO_2$ columns are removed from slant columns to retrieve the tropospheric $NO_2$ for Solar Zenith Angles (SZA) lower than 80°, see Dieudonné et al. (2013) for a detailed description of the SAOZ tropospheric $NO_2$ retrieval. The SAOZ dataset of tropospheric $NO_2$ measurements at Paris was used in different studies to relate $NO_2$ concentrations at the surface with integrated $NO_2$ column in the boundary layer (Dieudonné et al., 2013), to interpret ozone measurements (Klein et al., 2017) and the seasonal cycle of ozone gradient (Ancellet et al., 2020).

SAOZ tropospheric $NO_2$ columns are available at the SAOZ webpage (http://saoz.obs.uvsq.fr/SAOZ_tropo_Paris.html and /SAOZ_tropo_Guyancourt.html, last access on 1 January 2021). These data were daily averaged between 6 and 18 UT and between 11 and 14 UT for comparison with satellite observations.

#### 2.1.2 TROPOMI data

Tropospheric $NO_2$ columns retrieved by TROPOspheric Monitoring Instrument (TROPOMI) aboard Sentinel 5 Precursor (S5P) satellite (Veefkind et al., 2012) launched in October 2017 were also used to discriminate air masses above SAOZ instruments benefiting from the high spatial resolution of this instrument ($3.5 \times 7$ km$^2$ and $3.5 \times 5.5$ km$^2$ since August 2019). TROPOMI is a passive-sensing hyperspectral nadir-viewing imager, aboard a near-polar sun synchronous orbit satellite at an altitude of 817 km, with an overpass at 13:30 local time and practically daily global coverage.

Retrieval applied on TROPOMI data allows distinction between tropospheric, stratospheric and total $NO_2$ columns. The algorithm was adapted from DOMINO/TEMIS approach for OMI (Boersma et al., 2007, 2011) based on Differential Optical

Absorption Spectroscopy (DOAS) method to obtain slant column densities (SCD) of $NO_2$ that are assimilated to the TM5-MP Chemical Transport Model (CTM) to separate the SCD. The CTM runs using 0-12 h forecast meteorological data from European Centre for Medium-Range Weather Forecasts (ECMWF) correspond to the OFF-line product. Finally, each slant column is converted to vertical column using pre-calculated Air Mass Factor (AMF) look-up-tables. Detailed description can be found at TROPOMI webpage (http://www.tropomi.eu/data-products/nitrogen-dioxide).

Van Geffen et al. (2020) analyzed the uncertainties of SCD of TROPOMI and compared them to OMI –QA4ECV data (Boersma et al., 2018). They show a very good agreement over a remote Pacific Ocean sector with a correlation of 0.99 but with 5 % higher values than the OMI–QA4ECV ones. Verhoelst et al. (2021) compared $NO_2$ total, tropospheric and stratospheric columns with the data of ground-based instruments Pandora, Multi-Axis Differential Optical Absorption Spectroscopy (MAX-DOAS) and Zenith-Scattered-Light DOAS (ZSL-DOAS or SAOZ) distributed around the world. Observations from MAX-DOAS were used for tropospheric comparisons since they are sensitive to absorbers in the lowest few kilometers of the atmosphere (Hönninger et al., 2004). A negative bias of 23 to 37% is observed in the cases of clean to slightly polluted conditions. In the case of highly polluted areas, the bias can reach 51%.

TROPOMI tropospheric $NO_2$ columns have been widely used to estimate the reduction of $NO_2$ amounts linked to the lockdown in 2020 established in different countries to prevent the spread of COVID19 (e.g. Bauwens et al., 2020, Biswal et al., 2020, Ding et al; 2020, Koukouli et al.,2020, Lieu et al., 2020).

In his validation paper against consolidated ground-based data, Verhoelst et al., 2021 was using TROPOMI's tropospheric columns of $NO_2$ with a quality assurance value (QA) higher than 0.75 to remove cloudy scenes presenting cloud radiance fraction higher than 0.5, snow- or ice-covered scenes, and problems in the retrieval. In our study, we have decided to use a less restrictive threshold of 0.5 in order to enhance the number of days and to avoid biasing the results towards clear day conditions. This resulted in doubling the number of data taken into account. The monthly mean $NO_2$ tropospheric columns of TROPOMI present similar seasonal evolution within $2\sigma$ for both QA (not shown).

TROPOMI tropospheric $NO_2$ columns are available at Copernicus webpage (https://s5phub.copernicus.eu).

## 2.2 Surfaces concentrations

AIRPARIF is a network of standard in situ sensors to monitor air quality over Ile-de-France region. One of the key variables measured by AIRPARIF is $NO_2$. Hourly $NO_2$ concentrations are measured at most of the stations. The concentrations are measured by chemiluminescence (Fontijn et al., 1970) where the $NO_2$ amount is obtained after reduction to NO on a heated molybdenum converter. This kind of in situ sensor can overestimate ambient $NO_2$ concentrations due to interferences with non-$NO_x$ fraction of reactive nitrogen ($NO_z$). As an example, for urban sites in Mexico-city, Dunlea et al., 2007 found an average $NO_2$ overestimation for this type of sensor by 22%.

AIRPARIF network is formed by the 1) so-called "traffic" stations located at the edge of major traffic axes, 2) urban background stations, located in the city but not in the immediate vicinity of emission sources, 3) suburban and rural stations, and finally, a station installed on the top of the Eiffel Tower at an altitude of 300 m.

In this study, two AIRPARIF sites near the SAOZ of Paris were used, one considered as "traffic" site (Quai de Célestins) and the other as "urban" (Paris 13). AIRPARIF data of Versailles, nearest station to the SAOZ of Guyancourt was used to represent suburban site. Finally, two more stations at the foot (Paris 7) and on top of Eiffel Tower were considered to compare evolution of $NO_2$ concentration at different altitudes in the boundary layer. Data were obtained from Airparif webpage (https://www.airparif.asso.fr/telechargement/telechargement-station, last access on 22 January 2021). Daily average

data between 6 and 18 UT are used in this study as for SAOZ instrument

## 2.3 ERA-5 Reanalysis

ERA5 is the latest reanalysis of the ECMWF (European Centre for Medium-Range Weather Forecasts) generated by Copernicus Climate Change Service. ERA5 is produced by the Integrated Forecast System (IFS) CY41r2 version released in

2016 with a ten-member 4-D-Var assimilation each 12 hours. The horizontal grid resolution is ~31 km with 137 hybrid vertical levels up to 0.01 hPa (Hersbach et al., 2020). In addition to the significant increase of horizontal and vertical resolution of ERA5, as well as the 10 years' experience of model forecast and assimilation, new and reprocessed observational data records were considered. Further information can be found in online documents at ECMWF webpage (https://confluence.ecmwf.int/display/CKB/ERA5).

ERA 5 surface winds over Europe have been validated with wind observations from 245 stations in Europe, including two stations in Ile de France (Molina et al., 2021). The conclusion is that ERA5 is able to reproduce the wind speed from hourly to monthly time frequencies for any location in Europe with a Pearson's correlation coefficient varying from 0.6 to 0.85 in hourly scale and 0.9 to 0.95 in 24-hourly scale.

In this study, wind speed and direction at 950 hPa (mid-altitude of the convective boundary layer) were extracted from 0.25°

horizontal resolution in latitude and longitude data over the [48.75N, 49.00N], [2.00E, 2.50E] region at noon. The available quality-checked final product was considered for January 1st 2011 to October 31th 2020 and a provisional product for November-December 2020, the latter is expected to rarely differ from the final product (Hersbach *et al.,* 2020).

## 3 Methodology

The evaluation of lockdown effects on atmospheric $NO_2$ amounts is performed by selecting air masses moving from the

170 Parisian agglomeration to the suburban region. The objective is to consider only days when air masses for both sampling sites have a long enough residence time over the Paris area and have been influenced by local pollution. In this work, the sampling filter of air masses coming particularly from Parisian agglomeration was determined with the purpose of evaluating the decrease of human activities linked to the lockdown at Paris on both sites. The downwind direction from Paris to Guyancourt is privileged to filter out air masses originating from the western sector, which are mainly of oceanic origin, and

175 have only little encountered European emissions. Combined wind speed and direction are considered in this study to identify

such days. This procedure aims at selecting datasets with similar meteorological conditions for different years, so reducing the impact of interannual weather variability. The evolution of $NO_2$ concentrations and tropospheric columns at AIRPARIF and SAOZ stations (Table 1) are considered. The data of $NO_2$ concentration measurements by in situ instruments and $NO_2$ tropospheric column measurements by SAOZ were daily averaged between 6 and 18 UT. The measurements data are filtered using wind speed and direction of ERA5 analysis at noon to select weather conditions in which the Guyancourt site receives air masses that have passed the Paris agglomeration. Equation 1 represents the estimated residential time t of air masses coming from the center of Paris to Guyancourt.

$$t=\cos(\text{abs}(\text{dir\_g}-\theta_{era5})*\pi/180)*D/(v_{era5}), \hspace{4cm} (1)$$

where $v_{era5}$ and $\theta_{era5}$ correspond to speed and direction of wind at 12 UT and 950 hPa (altitude level in the middle of the convective boundary layer), dir\_g is the direction between Guyancourt and Paris (290°) and D is the approximate diameter of agglomeration (9.5 km) if we consider it as a circle.

Using this parameter t, three types of days were distinguished and for each class a linear fit between urban versus suburban observations was calculated:

1. Air masses of Parisian agglomeration not influencing Guyancourt or Versailles (t<0)

2. Air masses of Parisian agglomeration influencing Guyancourt or Versailles (t>0)

3. Air masses of Parisian agglomeration in a condition of weak wind influencing Guyancourt or Versailles, a subclass of the precedent one (t>30 min).

Figure 2 shows the scatter plot of SAOZ tropospheric $NO_2$ of Paris and Guyancourt (left panel) and AIRPARIF in situ $NO_2$ of Paris 13 district and Versailles (right panel) for the 2011-2020 period. Case 1 is represented by light green points, case 2 by grey points and case 3 by dark grey points. Linear orthogonal fit was applied for the three cases to highlight the relationship between urban and suburban stations for the different conditions of wind speed and direction. For each case, higher $NO_2$ amounts are observed at Paris, and the air masses at the surface present lower linear regression slopes than tropospheric columns. Case 1 presents the largest slopes, 2.99±0.01 (2σ standard error) for SAOZ measurements and 1.36±0.01 for AIRPARIF highlighting the importance of wind direction. In this case when Guyancourt is upwind of Paris, air masses pass over Guyancourt without having "touched" the agglomeration. Those air masses arriving in Paris center have crossed part of the agglomeration and then show larger $NO_2$ columns. Case 2 and 3 correspond to air masses generally crossing first the Parisian agglomeration and then south-west suburban region. They show slopes closer to unity. In case of SAOZ, the slopes of 1.38±0.01 and 1.31±0.01 were obtained for case 2 and 3, and the slopes of 1.11±0.01 and 1.04±0.03 in case of AIRPARIF, respectively. For our study, the classification of days with air masses associated to t>30 minutes will be considered, because in this case air masses pass over both stations with weak wind allowing for pollutant accumulation over the Paris agglomeration.

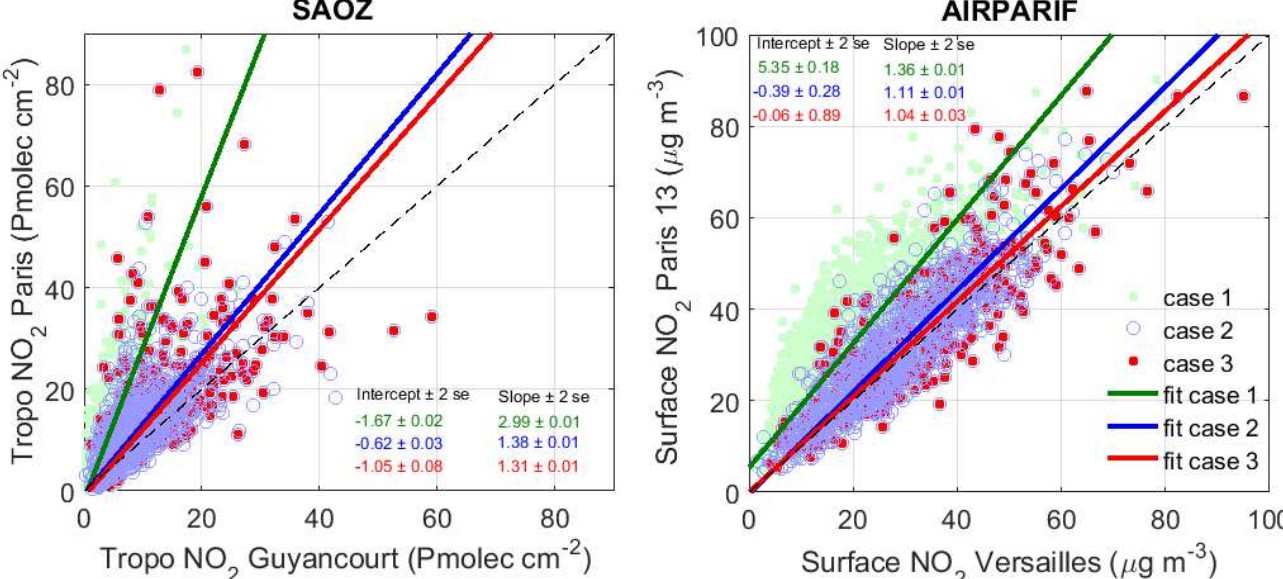

**Figure 2: Scatter plots of tropospheric (left panel) and surface (right panel) NO₂ measurements at Paris as a function of measurements at suburban station (Guyancourt and Versailles respectively) for different levels of t (see Eq. 1). Linear fits of the**
210 **different conditions are represented in green (case 1), blue (case 2) and red (case 3), see the text. The 1:1 line is represented by the black dash line. The estimated slope and it standard error is also shown for each case.**

The poorer correlation observed with SAOZ data could be explained since different types of air masses could be sampled at Guyancourt in the tropospheric column: those passing through the agglomeration center and accumulating NO₂ when passing
from the center to the edge (leading to larger columns at Guyancourt than at Paris), and those that have crossed only the limits of the agglomeration (leading to smaller columns at Guyancourt than at Paris).

## 4 Results

### 4.1 NO₂ evolution in 2020

The period preceding the lockdown represents meteorological conditions over Ile-the-France mainly characterized by high occurrence of oceanic air masses (see Fig. S3 of Petit et al., 2021) and fairly strong south-westerly winds (Fig. 3, left wind rose) preventing pollution events over this region. Changes in weather conditions three days after the implementation of lockdown on March 17th 2020 (middle wind rose on Fig. 3) were mostly anticyclonic contributed to the stagnation of pollutants in air masses advected from Paris to Guyancourt. Low wind speeds (<6m/s) are predominantly north-easterly in
the mid-March-to mid-May period. The period after the end of lockdown (Fig. 3, right wind rose) shows winds from south-westerly and north-easterly directions in the mid May to July period.

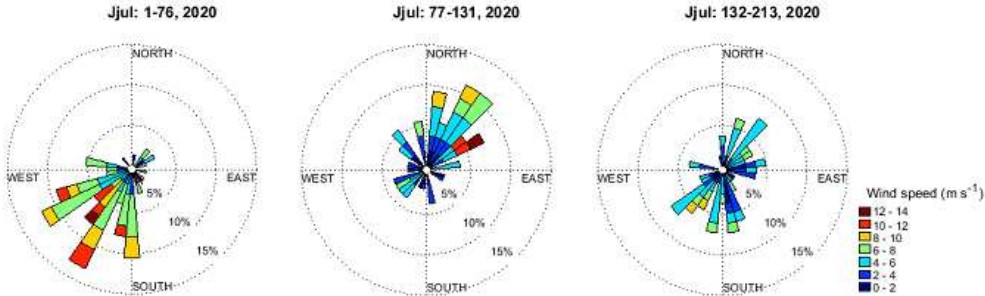

**Figure 3: From left to the right: wind rose from 12 UT ERA5 data before (1/1-16/3), during (18/3-10/5) and after (11/5-31/7) the 1st lockdown in France in 2020. The color indicates the wind speed in m s⁻¹. The color indicates the wind speed in m s⁻¹. The frequency**

**in % is showed by the circles.**

Figure 4 shows the evolution of tropospheric $NO_2$ columns in Paris (red curve) and Guyancourt (blue curve) in 2020 as observed by SAOZ (top panel). Colored points correspond to the filtered data with t>0 (open circles) and t>30 minutes (solid points). The filtered air masses at Paris and Guyancourt present similar values for most of the cases with coincident daily

events of increased tropospheric $NO_2$. Similar results are observed from in situ measurements at AIRPARIF stations (bottom panel). Vertical dashed lines are displayed in Figure 4 to separate 4 periods: before, during and after the lockdown and the last period of mixed restrictions (partial activities) since October 31th. The seasonal variability of $NO_2$ is well pronounced in the surface observations with a minimum in June and a maximum in winter.

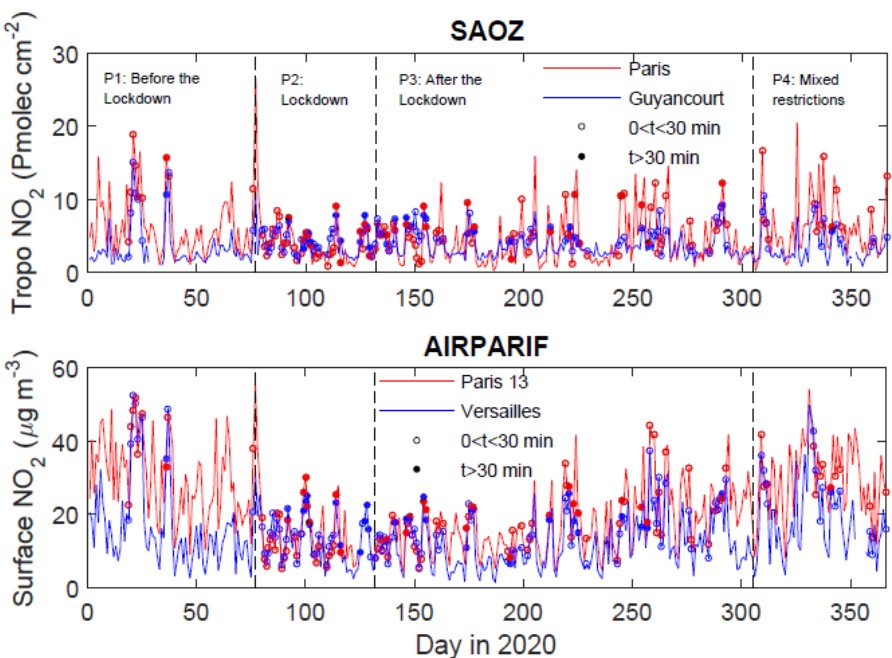

**Figure 4: Evolution of tropospheric $NO_2$ columns (top panel) and surface $NO_2$ (bottom panel) in 2020 at Paris and south-west suburban stations. Vertical lines correspond to the day of period change: 17/03, 11/5 and 31/10.**

Table 2 shows different periods in 2020 related to restrictions imposed by French government to limit COVID19 propagation. During period 1 (before the lockdown) only two particular events with high $NO_2$ values above both stations are detected at the same time (t>0 min) by SAOZ instruments (Jan. 19-25 and Feb. 5-6). These events are also highlighted in AIRPARIF data. Only one day with t>30 min is observed on Feb. 5th. Frequent occurrence of oceanic air masses with high precipitation and wind speed leads to advection of clean air masses above the Île-de-France region before lockdown period (Viatte et al., 2021) and low $NO_2$ values are observed, lower than observed during period 2 (lockdown) for suburban stations (Guyancourt and Versailles). A $NO_2$ peak is observed on March 17th coincident to the start of the lockdown period, which could be linked to the massive departure of Parisian inhabitants. A change of weather conditions in the beginning of period 2 with low north-easterly wind speeds promote the accumulation of polluted air masses over Île-the-France. Most of the days are characterized by a residential time t > 30 min. Despite this situation, levels of tropospheric $NO_2$ remain low; this certainly illustrates the decrease of emissions during the lockdown period. The period 3 (after the lockdown) started on May 11th 2020 and $NO_2$ values remained low until the second week of July (beginning of scholar holidays) with $NO_2$ enhancement events comparable to period 2. Since then, higher $NO_2$ values of pollution events are observed by SAOZ and AIRPARIF instruments showing slight differences between urban and suburban stations for days with t > 30 min. A less restrictive lockdown (open schools and less restrictive movement of people).

**Table 2. The four periods in 2020 shown in Figure 4 and the related restrictions imposed by the French government to limit the COVID19 propagation.**

|  | Periods in 2020 | Restrictions |
|---|---|---|
| P1 | 1 Jan to 16 March | Not any |
| P2 | 17 March to 10 May | 1st lockdown: non-essential stores, schools, cultural establishments, etc closed. Only displacements <1km and with a certificate are authorised. Teleworking is strongly suggested. |
| P3 | 11 May to 29 October | Gradual lifting of restrictions: schools and non-essential stores opened with imposed physical distancing and masks. Possible displacement without certificate. A curfew was imposed mid-October. Teleworking is still recommended. |
| P4 | 31 October to 15 December | 2nd lockdown: schools opened but universities still closed. Some activities are allowed: Some non-essential stores opened with strong restrictions. Some restrictions as displacement of 1km maximum are relaxed at the end of November. |

### 4.2 Comparison to previous years

### 4.2.1 Tropospheric $NO_2$ columns

TROPOMI tropospheric $NO_2$ measurements in 2020 were widely used to show a decrease of $NO_2$ amounts in different countries, which was attributed to policies restricting human activities by comparing lockdown and pre-lockdown period or same period in 2019 (e.g. Ding et al., 2020; Koukouli et al., 2020; Prunet et al., 2020; Siddiqui et al., 2020). SAOZ measurements between 11 and 14 UT were averaged to match overpass time of TROPOMI above the stations. TROPOMI

data was previously filtered for the qa>0.5 (see Sub-section 2.1.2) and a radius of 5 km around SAOZ stations. Figure 5 shows the evolution of the monthly mean and two standard error (2σ) of tropospheric $NO_2$ columns above Paris and

Guyancourt stations since January 2019 observed by SAOZ and TROPOMI (left panels). The standard error corresponds to the standard deviation of the mean divided by the root number of considered days. Similar inter-monthly evolution is observed by both instruments with a generally good agreement within ±2σ and a correlation of 0.80 at Paris and 0.70 at Guyancourt. TROPOMI presents generally lower $NO_2$ values than SAOZ but within the 2σ uncertainty level. This is not the case in May 2020 (month 17 on Fig. 5) during which TROPOMI $NO_2$ amounts are significantly larger at 2σ level than

SAOZ. Monthly mean values present a seasonal variation reaching values above 10 Pmolec cm$^{-2}$ in winter at Paris while they vary between 4 to 7 Pmolec cm$^{-2}$ at Guyancourt. The first months of 2020 present lower values compared to 2019, mostly due to weather conditions while March-May $NO_2$ decrease (month 15-17) is coincident with the lockdown period. A histogram of the differences between TROPOMI and SAOZ is also shown in Figure 5 (right panels). A mean and median difference of -0.2 Pmolec cm$^{-2}$ and +0.12 Pmolec cm$^{-2}$ respectively is obtained at Paris station and of -0.6 Pmolec cm$^{-2}$ and -

0.7 Pmolec cm$^{-2}$ respectively at Guyancourt. It corresponds to a median relative difference of 2% at Paris and -22% at Guyancourt stations. Dispersion of the difference represented by the half of the 68% interpercentile (IP68/2) is 2.9 and 1.6 Pmolec cm$^{-2}$ respectively at Paris and Guyancourt.

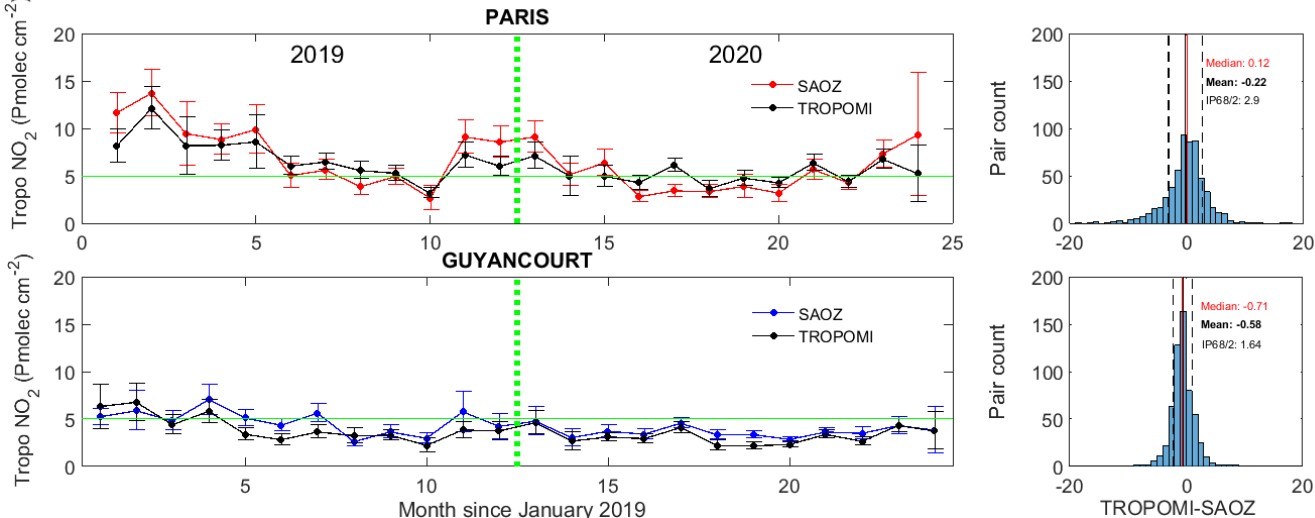

**Figure 5: Left panels: Monthly mean tropospheric $NO_2$ and 2σ standard error above Paris (upper panel) and Guyancourt (bottom**
**panel) measured by ground-based SAOZ instrument (color lines) and TROPOMI satellite instrument (black lines). Right panels:**
**Histogram of TROPOMI-SAOZ differences at Paris (upper panel) and Guyancourt (bottom panel). Vertical lines represent the**
**median, mean and dispersion by the half of the 68% interpercentile (IP68/2).**

TROPOMI and SAOZ data selected for days with t > 30 min were averaged between 11h and 14h UT for the period of the
2020 lockdown in France (March 17[th] to May 10) and median values were computed from the SAOZ and TROPOMI data

for the 2011-2020 annual range (Figure 6). TROPOMI $NO_2$ decrease in 2020 compared to 2019 is 35±12% for Paris and 22±27% for Guyancourt. Bauwens et al. (2020) have found a decrease of 28% during the 21st days of lockdown over 50 km region centered at Paris using TROPOMI and OMI data compared to same period in 2019. A larger tropospheric $NO_2$ decrease of about 47% is found from SAOZ observations between 2019 and 2020 at both studied stations (see Figure 6).

Prunet et al. (2020) found an even large decrease of $NO_2$ values varying from 52% to 86% during the lockdown in a 120 km region around Paris using yearly 2019-2020 TROPOMI data and the city-scale $NO_2$ plume mass method.

It should be noted that the SAOZ data sets show a long-term negative trend since 2011. Font et al. (2019) have used in situ data to study the impact of policy initiatives in different megacities. They have shown a mean $NO_2$ decrease on roadside (background) sites of -2.9 (1.7) % $yr^{-1}$ in Île-de-France for the 2010-2016 period, linked to the introduction of Euro V heavy-

duty vehicles regulations since October 2009. Others policies were implemented after then (e.g. Euro VI since 2014). The trend of tropospheric $NO_2$ amounts needs to be considered to better quantify the effects of lockdown on air pollution, which cannot rely on the comparison with a single reference year as was done in many other studies (e.g. Bauwens et al., 2020; Prunet et al., 2020).

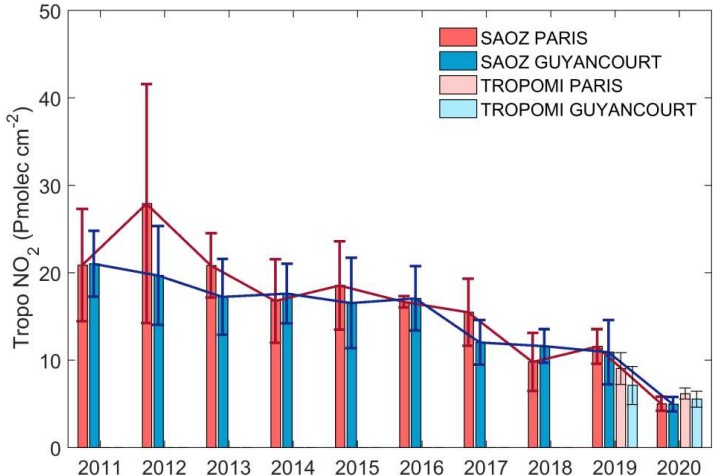


**Figure 6: Tropospheric $NO_2$ median values of March 17th – May 10th period at Paris and Guyancourt from SAOZ observations since 2011 and TROPOMI measurements in 2019 and 2020. Error bars represent 1σ.**

To better account for traffic-related pollution events in the daily averaged $NO_2$ columns the full daytime data of tropospheric

$NO_2$ measurements by SAOZ (SZA<80°) of the corresponding day were considered. The median value of daily columns with t>30 minutes was computed for each year during period 2 and 3 above Paris and Guyancourt. Period 1 and 4 were not considered since only one day with t >30 min was observed above the stations during these periods in 2020. Period 3 was restricted to May 11th - July 15th (period 3') to avoid the effect of $NO_2$ seasonal variation in the final median value. A robust regression fit (reweighted bisquare function to reduce weight of outliers far ~5 times from the median) was applied to period

2 and 3' to compute the trend for the 2011-2019 period. We will focus only on the period of lockdown since important $NO_2$ interannual variability in the period 3' does not present a $2\sigma$ significant slope value neither at Paris, nor at Guyancourt. Only the lockdown period presents a significant negative slope of $-1.51\pm0.48(1\sigma)$ Pmolec cm$^{-2}$ yr$^{-1}$ at Paris and $-1.42\pm0.14(1\sigma)$ Pmolec cm$^{-2}$ yr$^{-1}$ at Guyancourt as shown in Figure 7. These values correspond to a negative trend of $-5.86\pm1.92$ % yr$^{-1}$ at Paris and $-6.79\pm0.66$ % yr$^{-1}$ at Guyancourt relative to 2011. Previous studies have presented similar values over Western

Europe. Zhou et al., (2012) found significant negative trends in the 2004-2009 period varying from -4 to -8 % yr$^{-1}$ using OMI tropospheric $NO_2$ columns. Curier et al. (2014) computed the trend from the synergistic use of OMI $NO_2$ tropospheric columns and the chemistry transport model LOTOS–EUROS, finding significant negative trends of 5-6% yr$^{-1}$. The year 2020 presents the lowest values of $NO_2$ at both stations (5.4 Pmolec cm$^{-2}$ at Paris and 4.4 Pmolec cm$^{-2}$ at Guyancourt) that are significantly different at $1\sigma$ from previous years (Figure 7). The median value in 2020 is lower than the extrapolated value

using the computed 2011-2019 trend by $55.6\pm15.7\%$ at Paris and by $45.6\pm11.8\%$ at Guyancourt. If the tropospheric median column of $NO_2$ in 2019 had been used as a reference for comparison, slightly higher declines would have been obtained within $\pm1\sigma$: $56.7\pm9.1\%$ and $52.6\pm14.5\%$ at Paris and Guyancourt, respectively. Choosing other reference years would obviously yield different results, e.g. slightly lower value at Paris ($55\pm10.7\%$) and even higher at Guyancourt ($58.9\pm12.5\%$) when using year 2018 as a reference (Figure 7). Moreover, choosing earlier years as a reference would pose the problem of

$NO_2$ variability factors associated with both the lockdown and the long-term $NO_2$ reductions. This confirms the advantage of our method that calculates the reference from a decadal data base and corrects for the long-term trend. It should be noted that the data filtering procedure based on meteorological conditions (wind speed and direction) significantly changes the result of the $NO_2$ reduction estimate in Guyancourt, making it statistically insignificant ($9.7\pm41.6\%$) if filtering is not applied; at the same time the estimate for Paris has not changed much ($58.3\pm20.9\%$). Table 3 presents a summary of the NO2 reductions in

2020 using different datasets described previously in the text. This indicates that results at Paris site located in the center of the agglomeration are not dependent in 2020 on meteorological conditions. On the contrary, for the Guyancourt site at the edge of the agglomeration selecting the days when the site is impacted by emissions within the agglomeration is crucial.

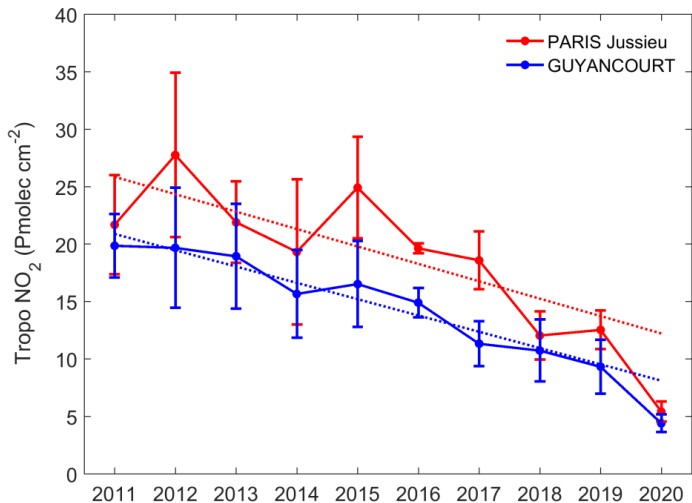

**Figure 7: Interannual variability of tropospheric NO₂ median values of March 17th – May 10th period at Paris and Guyancourt computed from SAOZ observations since 2011. Error bars represent 1σ standard error. Computed robust fit is shown by the dotted color lines.**

**Table 3: Dataset used to compute NO₂ reductions in 2020: instrument, time period in UT to calculate the daily mean value, the reference value and application of the filter of the residential time. The last columns correspond to the corresponding computed reductions in % for Paris and Guyancourt. Significant values at 1σ are in bold.**

| Dataset | Daily mean (UT) | Reference | Filter | Paris | Guyancourt |
|---------|-----------------|-----------|--------|-------|------------|
| TROPOMI | 11-14 | 2019 | Yes | **35** | 22 |
| SAOZ | 11-14 | 2019 | Yes | **47** | **47** |
| SAOZ | 6-18 | 2019 | Yes | **56.7** | **52.8** |
| SAOZ | 6-18 | 2018 | Yes | **55.0** | **58.9** |
| SAOZ | 6-18 | Trend in 2020 | Yes | **55.6** | **45.6** |
| SAOZ | 6-18 | Trend in 2020 | No | **59.3** | 9.7 |

### 4.2.2 Surface NO₂ concentrations

The annual median NO₂ concentration at AIRPARIF stations since 2011 (Table 1) were computed from daily available hourly data during the lockdown period filtered for the wind speed and direction as it has been done for the tropospheric NO₂ column (t>30 minutes). Figure 8 presents the interannual variability of NO₂ concentration at the five AIRPARIF stations. In addition, the calculated robust fit for the decadal evolution at each station is shown. The background or urban stations (Paris 7 and 13) present similar interannual variability with higher values at Paris 7. The station of Quai de Celestins in close proximity to local traffic shows much higher values, which are significantly different from those at other urban sites. The suburban station of Versailles presents similar values to Paris 13 at ±1σ. The observation station located at 300 m of the Eiffel Tower near Paris 7 station shows the lowest values.

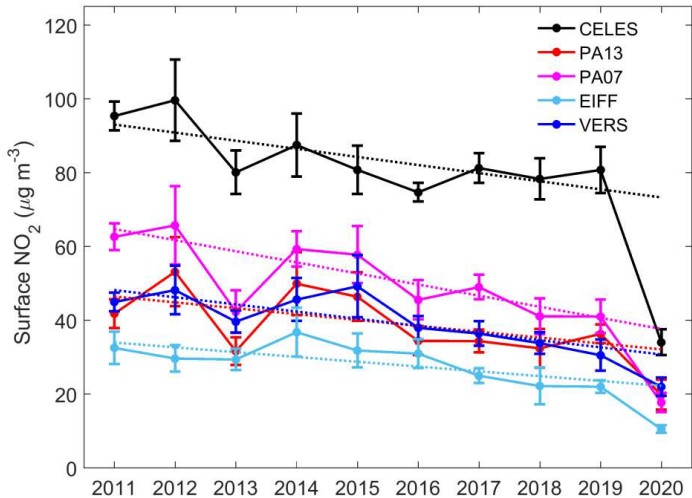

**Figure 8: Similar to Figure 7 but with surface NO₂ concentration for different in situ sensors of AIRPARIF network (see Table 1).**

The five AIRPARIF stations present negative trends from -3 to -1.3 µg m$^{-3}$ yr$^{-1}$ equivalent to -4.6 to 2.4 % yr$^{-1}$ (Table 4).
Font et al. (2019) found similar negative trend varying from -3.4 to -2.4 % yr$^{-1}$ for roadside stations at Paris for the 2010-2016 period. These trends appear to be less negative than those obtained from column measurements. Possible reasons for this are an increase of the NO₂ to NOx emission ratio, and a limitation by the available amount of O₃ for the NO to NO₂ conversion. Both factors affect more strongly the surface concentration than the boundary layer column, which could lead then to the different trend estimates.

Incomplete NO to NO₂ conversion is for example suggested by NO₂ and ozone concentrations of the same order of magnitude at Paris urban background sites (Figure 38 of Airparif 2019). In such a situation, the NO₂ trends are both impacted by the NOx emission and ozone trends. Figure 38 in Airparif (2019) cited above shows indeed strongly increasing ozone average urban background over Paris, for instance 35 to 43 µg m$^{-3}$ respectively for the 2007-2009 and 2017-2019 periods. This positive ozone trend buffers to some extent the negative NOx emission trend.

However while this reasoning would qualitatively explain differences in trends between column and in situ measurements, it fails to explain differences in trends between different in-situ sites, in the sense that larger NOx values would lead to smaller negative trends. This is not observed, on the contrary, the NO₂ trend is more negative at ground of Eiffel tower than at altitude when NOx becomes lower. Thus the exact explanation of differences in trends at different sites and heights still need more investigations. In 2020 significant decrease compared to the extrapolated value using the above calculated linear trends

is observed at all stations and reach similar median values, slightly higher for the traffic station and slightly lower for Eiffel Tower observation station. The relative values of NO₂ reductions are shown in Table 4. Comparable values at 1σ are observed for traffic and urban stations in Paris, with lower values at Paris 13 where standard error is higher. Nevertheless, the reduction of NO₂ concentration observed in absolute values is more important at traffic stations (as CELES) compared to

urban station (as Paris 7). The observation station installed at 300 m of Eiffel Tower presents 53% of reduction identical to
Paris 7, station located at the foot of the tower. The suburban station of Versailles presents the lowest reduction of 28.5%,
significantly different to other stations at $1\sigma$ except for Paris 13. It should be noted that both stations show an almost twice
larger standard deviation of 14%. Reasons for these lower values are not clear. It can be speculated that at this suburban site
the relative contribution of residential heating to $NO_x$ sources is stronger than at Paris sites, and probably these sources have
increased during the lockdown due to the presence of people at home (Menut et al., 2020).


**Table 4: AIRPARIF stations, type, NO₂ trend ±1σ in µg m⁻³ y⁻¹ and NO₂ reduction in 2020 compared to the estimated value as a function of the computed trend.**

| Station | Type | Trend (2011-2019) ± 1σ (µg m⁻³ yr⁻¹) / (% yr⁻¹) | Reduction in 2020 ± 1σ (%) |
|---------|------|-----------------------------------------------|----------------------------|
| CELES | TRAFFIC | -2.19±0.85 / 2.36±0.92 | 53.6±5.4 |
| PA13 | URBAN | -1.59±1.04 / -3.34±2.25 | 38.3±14.6 |
| PA07 | URBAN | -3.01±0.81 / -4.65±1.25 | 52.9±8.4 |
| EIFF | OBSERVATION | -1.30±0.51 / -3.83±1.49 | 52.8±9.4 |
| VERS | SUBURBAN | -1.94±0.58 / -4.02±1.18 | 28.5±13.1 |

Collivignarelli et al. (2021) compared the NO₂ concentration observed by the traffic and urban stations of AIRPARIF during
the lockdown in 2020 to the same period in previous years (2017-2019). They found a decrease of 15% for urban and 33%
for traffic stations. However, when considering similar meteorological conditions with respect to rainfall, temperature and
wind speed, the authors found a reduction of 51.5 % corresponding to traffic stations and approximately 45% for background
ones, similar to values obtained in this study.

**5 Discussion**

Various studies have been conducted to assess the impact of recent lockdowns on air quality in many countries around the
world due to COVID-19 pandemic. In a number of works, the observed NO₂ contents were compared with respective levels
for the same period of previous years using ground-based and/or satellite measurements. Shi and Brasseur (2020) found a
decrease of NO₂ concentrations in China by 50% compared to 2019 during the same period of the lockdown and by 60%
compared to 2018, highlighting the interannual variability of NO₂ reductions that could depend on meteorological conditions
or long-term variability. Others authors compared NO₂ amounts before and during lockdown. For example, Siddiqui et al.
(2020) observed 46% reduction of NO₂ tropospheric columns in India using satellite data, Liu et al. (2020) estimated 48% of
reduction in China before and during the Lunar New Year, which is 21% more than in previous years 2015-2019 (given that
a NO₂ reduction has been observed over the past years even without COVID); Bauwens et al. (2020) deduced 20-38%
reduction in Western Europe. Many studies have considered specific techniques to limit the effect of meteorological
conditions in their data. In the case of Paris, 45-52% reduction of NO₂ concentration was estimated by Collivignarelli et al.

(2021) using equivalent temperature and wind speed days, ~50% by Barré et al. (2020) using a Gradient Boosting Machine Learning (GBML) technique. In case of tropospheric $NO_2$ columns measured by satellite instruments, Prunet et al. (2020) estimated a 2 weeks averaged reduction of $NO_2$ varying between 52 and 86% using the city-scale $NO_2$ plume mass method for March 16th-April 26th. In the present study, the long-term evolution was considered from one decade of measurements combined to air masses filtering based on slow wind speed and long residence time. The calculated reductions in the tropospheric $NO_2$ column and surface concentration are comparable in magnitude to the results of previous studies in Western Europe: 46-56% and 28-54%, respectively.

Menut et al. (2020) compared the results of two special model calculations performed for the March 2020 lockdown period in Western Europe. They used the WRF-CHIMERE model for two simulations: one using Business As Usual (BAU) scenario with classical emissions and the other one using realistic scenario taking into account an estimate of lockdown measures on $NO_2$ in 2020. The authors found a maximum reduction of 43% of average $NO_2$ concentration over France. This simulation was based on a reduction in emissions of about 80% in the transport sector and 40% reduction in the industrial sector, but an increase for residential emissions during the second half of March, reducing emissions of $NO_x$ probably by more than 50% (taking into account the distribution of $NO_x$ emissions as given by CITEPA (https://www.citepa.org/fr/2020-nox/). Thus, $NO_2$ concentration reductions are slightly lower than $NO_x$ emissions changes in these simulations, probably due to an increase in the $NO_2/NO$ ratio for lower $NO_x$ concentrations. This suggests that, at least when spatially averaged, $NO_x$ emission reductions due to lockdown are similar to those of $NO_2$ surface concentrations.

**6 Conclusions**

To assess the impact of France's policy decision to limit the spread of the SARVS-CoV-2 virus by establishing a restrictive lockdown between March 17 and May 10, 2020, $NO_2$ surface concentrations and tropospheric columns over Île-de-France were analyzed, more specifically in Paris and suburban areas in the south-west of the agglomeration. Possible factors that can influence $NO_2$ changes other than $NO_x$ emissions reduction due to lockdown were considered. The data sets were partitioned to select the conditions of light winds moving air masses from Paris to a suburban area in the southwest. In addition, the known long-term reduction of $NO_2$ is also considered using the measurements in the previous decade. The tropospheric $NO_2$ reduction obtained from the SAOZ data is about 50% (56% at Paris site and 46% at the southwest suburban site). These values are close to the literature data found for Europe within the estimated error bars (Barré et al., 2020; Prunet et al., 2020). This work highlights the ability of satellite TROPOMI measurements to distinguish between urban and suburban sites tropospheric columns, showing higher mean values at an urban station compared to a suburban one. The latter is also confirmed by the ground-based SAOZ measurement data. The agreement between the evolution of $NO_2$ in the troposphere observed at urban and suburban sites improves when selecting similar meteorological conditions. Surface $NO_2$ concentrations inside Paris are highly influenced by local pollution and differences between the data of traffic and background urban sites are observed as expected. Surface concentrations were reduced by ~50% at all stations (similar at

±1σ), except the site of Paris 13 in the Choisy Park that shows a lower reduction. The suburban station of Versailles presents $NO_2$ concentrations similar to Paris 13 and the reduction in 2020 was 10% lower, within the error bars.

The reductions at Paris sites during the lockdown are important using or not a filter to remove the effect of different meteorological conditions. On the contrary, selecting data according to air mass residence time over the agglomeration, strongly changes the estimates of $NO_2$ reductions at the suburban sites. As expected, if filtering is not applied, lower $NO_2$ reductions are found for suburban sites, since the datasets include also measurements that are less affected by the agglomeration and closer to background conditions. If the long-term evolution is not considered, the computed reductions

highly depend on the year of reference. In this study, a negative tropospheric $NO_2$ trend of -1.5 Pmolec $cm^{-2}$ $yr^{-1}$ (equivalent to ~6.3 % $yr^{-1}$) is observed. Surface $NO_2$ concentrations also show negative trends with a mean value of -2.2 μg $m^{-3}$ $yr^{-1}$ (~3.6 % $yr^{-1}$).

In conclusion, the negative trend estimated during the last decade, indicates the long-term benefits of the environmental measures taken to reduce $NO_x$ emissions. The magnitude of the $NO_2$ supplementary reduction in 2020, which we calculate to

be around 50%, is consistent with the reduction in emissions associated with the lockdown in France, as suggested in a recent modelling study (Menut, 2020).

*Data availability*. The data used in this study are publicly available: tropospheric $NO_2$ from SAOZ instruments on http://saoz.obs.uvsq.fr, last access: 1[st] January 2021 and from TROPOMI satellite instrument on

https://s5phub.copernicus.eu, last access: 3[rd] January 2021; $NO_2$ concentrations from AIRPARIF on https://www.airparif.asso.fr/telechargement/telechargement-station, last access: 22 January 2021 and wind speed and direction from ERA5 on https://confluence.ecmwf.int/display/CKB/ERA5, last access: 20 January 2021.

*Author contributions*. AP, FG and MP contributed to the processing, analysis and availability of SAOZ data. AB and DI

processed the TROPOMI data. AH provided ERA5 data above Paris. MB developed the filter method to account for meteorological conditions. AP and SGB performed the statistical analysis. AP wrote the paper with the assistance from all authors.

*Competing interests*. The authors declare that they have no conflict of interest.


*Acknowledgements*. The authors warmly thank the French Institut National des Sciences de l'Univers (INSU) of the Centre National de la Recherche Scientifique (CNRS) and the Centre National d'Etudes Spatiales (CNES) for supporting observations of SAOZ instruments of the French Research Infrastructure ACTRIS. The SAOZ instrument of Paris is hosted at QUALAIR platform in Sorbonne University and the one of Guyancourt at SAOZ Unit/ACTRIS platform in UVSQ

University. We thank the Copernicus Services Data Hub for providing the TROPOMI/S5P data, AIRPARIF for the in situ observations and ECMWF for ERA5 wind data. This project was supported by ValS5PSAOZ project of CNES.

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
