# Peer review of "Impact of COVID-19 pandemic related to lockdown measures on tropospheric NO2 columns over Île-de-France"

_Atmospheric Chemistry and Physics, 2021_

## Author Comment (AC1)

**Reply to Anonymous Referee #1 review of manuscript acp-2021-456**

**Impact of COVID-19 pandemic related to lockdown measures on tropospheric NO2 columns over Île-de-France**

Andrea Pazmino on behalf of all co-authors

We warmly thank Anonymous Referee #1 for the interest that he showed in our work and for the time spent on its evaluation. Your valuable comments have helped us to improve our manuscript. Please find our answers to your comments (in red)

*This study addresses the influence of physical distancing, due to the COVID-19 pandemic, in NO2 concentration over Île-de-France. The manuscript has very interesting results and performs a good comparison with similar studies.*

**Main questions**

*Line 117: Please, detail or add some reference about this quality assurance.*

Recently Verhoelst et al., 2021 validated total, stratospheric and tropospheric columns of $NO_2$ of TROPOMI against consolidated ground-based data. In the case of tropospheric $NO_2$, the TROPOMI's quality assurance value (QA) higher than 0.75 is used to remove cloudy scenes presenting cloud radiance fraction higher than 0.5, snow- or ice-covered scenes, and problems in the retrieval. In our study, we have decided to use a less restrictive threshold of 0.5 to enhance the number of days taken into account and not to bias the results to clear day conditions. In order to evaluate the impact on SAOZ and TROPOMI comparison using a less restrictive QA, the monthly mean tropospheric $NO_2$ above Paris of TROPOMI was computed considering only data with QA>0.75 (Figure 1, bottom panel) and results of QA> 0.5 of the paper were included in the upper part of the figure. Only SAOZ coincident days with TROPOMI are taken into account to compute the monthly mean.

[Figure]

Figure 1: Left panels: Monthly mean tropospheric $NO_2$ and $2\sigma$ standard error above Paris measured by ground-based SAOZ instrument (red lines) and TROPOMI satellite instrument (black lines) with QA>0.5 (upper panel) and QA>0.75 (bottom panel). Right panels: Histogram of TROPOMI-SAOZ differences for TROPOMI QA >0.5 (upper panel) and QA>0.75 (bottom panel). Vertical lines represent the median, mean and dispersion by the half of the 68% interpercentile (IP68/2).

A similar evolution of tropospheric $NO_2$ is observed using QA>0.5 or 0.75. Approximately twice as much TROPOMI data is considered for QA>0.5 than QA>0.75. The median value of the difference is of the same order of magnitude and the dispersion is slightly higher for QA>0.5.

The comparison of the two TROPOMI datasets is presented in Figure 2. The monthly mean values present similar seasonal evolution within 2σ except on December 2020 where only one value is observed for QA>0.75.

[Figure]

Figure 2: Monthly mean tropospheric $NO_2$ and 2σ standard error above Paris measured by TROPOMI satellite instrument with QA>0.5 (black lines) and QA>0.75 (blue lines)

As a conclusion of this discussion, we decided to keep the TROPOMI data with a QA above 0.5 for this study.

The following paragraph was removed at the end of Section 2.1.2.

*"The data have been filtered using the quality assurance value higher than 0.5."*

and replaced by

*In his validation paper against consolidated ground-based data, Verhoelst et al., 2021 was using TROPOMI's tropospheric columns of $NO_2$ with a quality assurance value (QA) higher than 0.75 to remove cloudy scenes presenting cloud radiance fraction higher than 0.5, snow- or ice-covered scenes, and problems in the retrieval. In our study, we have decided to use a less restrictive threshold of 0.5 in order to enhance the number of days and to avoid biasing the results towards clear day conditions. This resulted in doubling the number of data taken into account. The monthly mean $NO_2$ tropospheric columns of TROPOMI present similar seasonal evolution within 2σ for both QA (not shown).*

*Is there some previous validation of ERA-5 data over Île-de-France?*
ERA 5 surface wind over Europe have been validated with wind observations from 245 stations in Europe, including two stations in Ile de France (Molina et al., 2021). The conclusion is that ERA5 is able to reproduce the wind speed from hourly to monthly time frequencies for any location in Europe with a Pearson's correlation coefficient varying from 0.6 to 0.85 in hourly scale and 0.9 to 0.95 in 24-hourly scale.

Reference
Molina, M. O., Gutiérrez, C., and Sánchez, E.: Comparison of ERA5 surface wind speed climatologies over Europe with observations from the HadISD dataset, Int. J. Climatol., 1–15, joc.7103, https://doi.org/10.1002/joc.7103, 2021.

The following phrase and the corresponding reference was introduced in L142 of the paper
*ERA 5 surface wind over Europe have been validated with wind observations from 245 stations in Europe, including two stations in Ile de France (Molina et al., 2021). The conclusion is that ERA5 is able to reproduce the wind speed from hourly to monthly time frequencies for any location in Europe with a Pearson's correlation coefficient varying from 0.6 to 0.85 in hourly scale and 0.9 to 0.95 in 24-hourly scale.*

*Is it possible to add information about the variation of physical distancing rate observed during the four stages (P1, P2, P3, and P4)? Such information can help the discussion presented in sections 4.1 and 4.2.*
The following phrase was added in L203 as well as the Table
"Table 2 shows different periods in 2020 related to restrictions imposed by French government to limit COVID19 propagation.

Table 2. The four periods in 2020 shown in Figure 3 and the related restrictions imposed by the French government to limit the COVID19 propagation.

| Periods in 2020 | | Restrictions |
|---|---|---|
| P1 | 1 Jan to 16 March | Not any |
| P2 | 17 March to 10 May | 1st lockdown: non-essential stores, schools, cultural establishments, etc closed. Only displacements <1km and with a certificate are authorised. Teleworking is strongly suggested. |
| P3 | 11 May to 29 October | Gradual lifting of restrictions: schools and non-essential stores opened with imposed physical distancing and masks. Possible displacement without certificate. A curfew was imposed mid-October. Teleworking is still recommended. |
| P4 | 31 October to 15 December | 2nd lockdown: schools opened but universities still closed. Some activities are allowed: Some non-essential stores opened with strong restrictions. Some restrictions as displacement of 1km maximum are relaxed at the end of November. |

*The variation of wind speed and direction is a relevant factor to demonstrate the pollutants dispersions. However, to provide a better discussion about the meteorological influence more parameters could be presented like temperature, rainfall rate, the occurrence of thermal inversions, atmospheric boundary layer height.*
It is true that many others meteorological parameters could have at least an indirect impact on $NO_2$ columns. Nevertheless, we decided to restrict our study to the main influencing parameters such as wind speed and direction, for which a direct physical relationship with $NO_2$ column densities can be established. For instance, boundary layer height does not directly affect $NO_2$ column densities, as they are by definition invariant against vertical mixing. In addition, the impact of precipitation on $NO_2$ is expected to be lower than for highly soluble compounds ($SO_2$, $PM_{10}$). The impact of temperature is more indirect, as a tracer of different air mass types (continental versus oceanic). Even if some statistical relationships may be established with these parameters, we preferred here to restrict to the much more straightforward wind related parameters.

*How were the different characteristics of each season considered? Could they have been responsible for the variations in the values found?*
In this study, only the first major lockdown period during mid-March and mid-May (Mach 17th-May 10th) called P2 was analysed quantitatively for deducing differences in $NO_2$ columns with respect to a reference period. Since this period is shorter than a season, seasonal variation was neglected.

*Line 142. As different seasons are considered, why the mid-altitude of the convective boundary layer was considered always as 950hpa?*

As explained before, only P2 restrictive lockdown period was considered and the mid-altitude of convective BL is appropriate within this period. This choice only affects the height level for which the wind data are taken from meteorological analysis. This height level needs to be located somewhere within the convective boundary layer. In Figure 3 (Figure 3.5 of Dieudonné, 2012), the time series of daily maximum BL height between July 2009 and February 2011 calculated from Rayleigh lidar measurements at Qualair station in Paris, co-located to the SAOZ instrument are shown. The BL values were selected considering only clear (red points) and cloudy sky days.  The black line corresponds to the rolling 30-day average. The figure shows clearly the variation of the BL over Paris as a function of season. These measurements are only available for a limited period in 2009-2011. They confirm the variation of the boundary layer as a function of season between 1.5 and 2.5 km, our choice seems appropriate. Within the convective boundary layer, wind variations are smaller than for example close to the surface. Choosing the height level at the exact middle of the boundary layer would much complicate the analysis and make results less transparent than choosing a fixed height level.

[Figure]

Figure 3. Time series of daily maximum Boundary layer height observed by the lidar of the Qualair station in Paris, with clear / cloudy days in red / blue. The black line represents the rolling 30-day average (Dieudonné, 2012).

Reference
Elsa Dieudonné. Analyse multi-instrumentale de l'influence de la variabilité de la hauteur de couche limite sur la distribution verticale des oxydes d'azote en région parisienne. Physique Atmosphérique et Océanique [physics.ao-ph]. Université Pierre et Marie Curie - Paris VI, 2012. Français. tel-00807665

**Technical questions**

*Figure 2: Please, add the units in the legend.*
The figure 2 was changed as follows:

[Figure]

Figure 2: From left to the right: wind rose from 12 UT ERA5 data before (1/1-16/3), during (18/3-10/5) and after (11/5-31/7) the 1[st] lockdown in France in 2020. **The color indicates the wind speed in m s$^{-1}$. The frequency in % is showed by the circles.**

*Line 228: "11 and 14 UT"*
Done

*Figure 4: Please, use the same labels in the x-axis of the upper and lower panel.*
Done

*Line 248: "11 and 14 UT"*
Done

---

## Author Comment (AC2)

**Reply to Anonymous Referee #2 review of manuscript acp-2021-456**

**Impact of COVID-19 pandemic related to lockdown measures on tropospheric NO2 columns over Île-de-France**

Andrea Pazmino on behalf of all co-authors

We thank Anonymous Referee #2 for the time devoted to evaluate our work. Your valuable comments have helped us to improve our manuscript. Please find our answers to your comments (in red)

*The topic of the manuscript (effects of lockdown measures on atmospheric NO2 levels) fits the scope of ACP. The manuscript is mostly well written and gives a clear description of the study with maybe a few more details needed on the methodology. The measurements reported and the analysis done is interesting and leads to reasonable results. However, there are two major questions that the authors need to answer convincingly before this study can be published.*

**Major comments**

*The main question to be answered is: What is new in this study? There have been hundreds of studies on Covid impacts on tropospheric NO2, using all kinds of instrumentation, and several of them even cover in-situ and TROPOMI measurements over Paris. To make this manuscript relevant, it needs to add new information and conclusions on the existing knowledge, and to me, it was not really clear what the new aspect of this study is. Please make this very clear in the revised manuscript.*

We agree with the reviewer that many studies were already done using $NO_2$ data. However, we are convinced that our study presents new original aspects, for the following reasons:

1. An original aspect of our study is to use a set of three different instruments for the analysis, allowing us to distinguish between the lockdown impact at surface and at rather local scale with in situ instrumentation, and more spatially integrated impacts affecting probably a large part of the agglomeration with tropospheric column measurements by the DOAS-Zenith Sky SAOZ instrument and by TROPOMI. While TROPOMI data have been used already in several studies, the DOAS measurements are available most of the time over the usual morning and afternoon traffic peaks (www.sytadin.fr), which puts the analysis on a statistically more secure basis increasing the time in the day to sample the pollution events. Differences in choosing different daytime periods are presented in Table 2 of the paper. They show a larger reduction of 6-10% at both sites in 2020 when the daytime period is larger by twice the time slot considered for TROPOMI intercomparison..

2. The use of two SAOZ instruments located at 24 km apart gives the possibility to distinguish the lockdown induced $NO_2$ evolution over an urban and a suburban site characterizing this perturbation in the context of the last decade. The same holds for the use of TROPOMI due to its high spatial resolution ($3.5 \times 7$ km2 and $3.5 \times 5.5$ km$^2$ since August 2019) data even if the data are only available from 2019 on.

3. The fact that the SAOZ instruments provide a long measurement time series over near a decade is another original aspect in this work. It avoids taking measurements in the spring period of a specific year as reference, which yields in general different results as mentioned in

the paper (table 2) for Guyancourt station showing ~59% and ~53% decrease using as reference year 2018 and 2019, respectively. Results presented in this paper are with this respect more robust. The paper thus provides error bounds for studies not being able to rely on an extended reference period as those for example using only TROPOMI satellite measurements.

4. Choosing a decadal reference period makes it necessary to compute the $NO_2$ trends during this time. Providing these trends is an important by-product of the paper. $NO_2$ column trends in Paris and surroundings are to our knowledge not available in the literature for the given period.

The paragraph L46 to L53 was modified in the introduction to better highlight the originality of our work:

*The objective of this study is to quantify the effect of $NO_2$ decreases due to lockdown considering long-term variability and meteorological conditions over Ile-de-France region during the last decade using different datasets characterizing the lockdown impact at local scale with in situ instrumentation, and at larger scale including a large part of the agglomeration with tropospheric column measurements. Two complementary sites are used, one in the center of Paris and the other one in the peripheral zone to highlight the possibly heterogeneous impact of lockdown in Ile de France region. The originality of the study is to rely not only on a single reference year before the COVID-19 pandemic that could strongly bias the study, but on a long decadal data set, in order to account for $NO_2$ variability on a longer period. Specific data filtering using wind speed and direction is applied in order to isolate data, which are affected by local pollution in the Greater Paris area, and to consider the changes in meteorological conditions for the different years.*

*The second point I'm struggling with is the logic behind the choice of wind directions for the two groups of stations (Paris centre and background). If I have understood the approach right, situations are selected for which Guyancourt is downwind of Paris, but why is that a good choice? I would have understood a selection where such wind directions are excluded in order to contrast city and background values, but this is not what the authors did. I really fail to see what the authors are trying to achieve with this set-up. Please explain the motivation for this choice and what we can learn from this particular set-up.*

The suggestion to exclude from the analysis the days at Guyancourt that are not affected by Paris air masses is interesting but our choice was exactly the opposite. We want to analyse only the days when Paris influenced Guyancourt to have similar characteristics of the air masses, especially in 2020 where the idea was to characterize particularly the abrupt decrease of traffic, mostly influenced by activities in the Greater Paris agglomeration. Looking at the background for air masses originating in the western sector did not seem particularly interesting to us, since these air masses are mainly of oceanic origin, and only little encountered European emissions. In Paris, we sample the center of the agglomeration, but in Guyancourt, we sample in addition air masses that have crossed only the periphery of the city, in particularly the south-west of the agglomeration.

The following figure shows the case 3 for SAOZ instruments when Guyancourt is downwind (red points) and upwind (blue circles) of Paris. The differences between urban and suburban stations are higher when air masses are coming from Guyancourt with a slope of 3.5 compared to 1.3 for the downwind case 3 used in our work.

[Figure]

Figure 1. Scatter plot of tropospheric NO$_2$ measurements at Paris as a function of measurements at suburban station of Guyancourt for case 3 (t>30 minutes) when Guyancourt is upwind (blue points) and downwind (red points) of Paris. Linear fits of the different conditions are represented in respective color. The 1:1 line is represented by the black dash line.

The following paragraph was added in the Section 3 (Methodology), L149 of the paper

*In this work, the sampling filter of air masses coming particularly from Parisian agglomeration was determined with the purpose of evaluating the decrease of human activities linked to the lockdown at Paris on both sites. The downwind direction from Paris to Guyancourt is privileged to filter out air masses originating from the western sector, which are mainly of oceanic origin, and have only little encountered European emissions.*

**Minor Comments**

*Line 39: Not clear what these percentages refer to*
The percentages refer to the fraction of global NOx emissions due to different major sources.
By rethinking about this sentence, we preferred to replace it by a statement about NOx emissions over greater Paris region, which is more relevant for the present study. So the new sentence reads:

*NOx levels are directly linked to human activities, for example over the Ile-de-France region, in which the Greater Paris region is imbedded, and for the year 2018, road traffic contributes to 53% of NOx emissions, followed by industry (13%, including also energy and waste treatment), residential heating (11%) and airports (9%) (https://www.airparif.asso.fr/surveiller-la-pollution/les-emissions, last consulted in August 2021).*

*Table 1: Please add a map with the locations for those not so familiar with the geography around Paris*

Here below the map with the SAOZ and AIRPARIF stations that will be included in the paper

[Figure]

Figure 2. Locations of the AIRPARIF (red points) and SAOZ (blue points) stations. Black dash line corresponds to the distance between both SAOZ stations. Map data © OpenStreetMap contributors under the license ODbL

*Line 100: In the discussion of the TROPOMI NO2, it would be good to also add a reference to van Geffen et al., 2020*
The following paragraph was added in L107
*Van Geffen et al. (2020) analyzed the uncertainties of SCD of TROPOMI and compared them to OMI –QA4ECV data (Boersma et al., 2018). They show a very good agreement* over a remote Pacific Ocean sector *with a correlation of 0.99 but with 5 % higher values than the OMI–QA4ECV ones.*

*Line 114: Typo Bawens*
Corrected to Bauwens

*Line 132: The statement about 24-hour averages is contradicted on the next page and if used, it should be explained why as this is then a different sample than the SAOZ measurements which do not cover night observations.*
Thank you for this remark. We indeed made a mistake. The 24-hour average is not used since only the same day period is used for SAOZ and in-situ measurements. This paragraph was corrected as follows
*Daily average data between 6 and 18 UT are used in this study as for SAOZ instrument*

*Line 135: last => latest*
Done

*Figure 1: I would suggest using the same range for x- and y-axis in the left panel, to include the 1:1 line, to use consistent colours for fitting line and points (green, blue, red) and to provide numerical values for slope and RMS*
We changed the figure and legend as suggested by the referee for more clarity.
The orthogonal regression function was applied to calculate the linear fit (see our answer to your following question). The following phrases were changed to consider the new values
L172-L173
*Case 1 presents the largest slopes, **2.11±0.02** (2σ standard error) for SAOZ measurements and **1.36±0.01** for AIRPARIF highlighting the importance of wind direction.*
L176-L177
*In case of SAOZ, the slopes of **1.38±0.01** and **1.31±0.01** were obtained for case 2 and 3, and the slopes of **1.11±0.01** and **1.04±0.03** in case of AIRPARIF,*

[Figure]

Figure 3. Scatter plots of tropospheric (left panel) and surface (right panel) $NO_2$ measurements at Paris as a function of measurements at suburban station (Guyancourt and Versailles respectively) for different values of t (see Eq. 1). Linear fits of the different conditions are represented in green (case 1), blue (case 2) and red (case 3), see the text. **The 1:1 line is represented by the black dash line. The estimated slope and it standard error is also shown for each case.**

*Figure 1: Why was the fit forced through zero – I could imagine that there is a higher NO2 background in the city centre*

Following the referees remark, we do not anymore force the graphs by zero. In addition, we now use an orthogonal regression function taking into account errors of the x and y variables, that is more adequate instead of a classical regression function that takes into account only the errors of the y variable. Doing so, we obtain small negative residuals for both for $NO_2$ columns and surface measurement, to which we do not attach any physical meaning (and individual observations are always positive!). Only for the case 1 of Airparif surface measurements we get a larger positive residual ($\sim +5$ µg m$^{-3}$) meaning that when $NO_2$ is zero at peripheral and probably upwind Versailles, it is still positive at the urban site (green line on right panel of Fig. 3).

[Figure]

Figure 4. Scatter plot of tropospheric $NO_2$ measurements at Paris as a function of measurements at suburban station of Guyancourt for case 3 (red circles) and case 3 filtered from weekend days (purple points). Linear fits of the different conditions are represented in respective color. The 1:1 line is represented by the black dash line.

*Figure 1: Why is the correlation for SAOZ so much poorer than for the in-situ observations? I would have expected the opposite – columns should be more conserved during transport than surface values. Please discuss.*

This is an interesting remark. We first looked at the other two pairs of surface sites and found again larger correlations than for the columns between Paris and Guyancourt (from 0.66 to 0.87 for AIRPARIF and 0.57 for SAOZ. The relatively weak correlation for SAOZ data is indeed astonishing. Differences are beyond the instrumental (retrieval) uncertainty which is estimated around 15-20%. An explanation for the lower correlation could then be that at Guayncourt we sample different types of air masses, those passing through the agglomeration center and accumulating $NO_2$ when passing from the center to the edge (leading to larger columns at Guyancourt than at Paris), and those that have crossed only the limits of the agglomeration (leading to smaller columns at Guyancourt than at Paris).

The following phrases were added to mention the poorer correlation for SAOZ data after the figure of the scatter plot of Urban Suburban stations

*The poorer correlation observed with SAOZ data could be explained since different types of air masses could be sampled at Guyancourt in the tropospheric column: those passing through the agglomeration center and accumulating $NO_2$ when passing from the center to the edge (leading to larger columns at Guyancourt than at Paris), and those that have crossed only the limits of the agglomeration (leading to smaller columns at Guyancourt than at Paris).*

[Figure]

Figure 5. Similar to Figure 1 for the in-situ traffic station of Quai des Celestins (left panel) and urban station of Paris 07 (right panel) as a function of suburban station of Versailles.

*Line 197: bleu => blue*
Done

*Line 228: Please provide more details on how the TROPOMI data were selected – which radius, which quality filter? How were the errors computed for TROPOMI and for SAOZ?*
Details the referee asked for were added to this paragraph:
*SAOZ measurements between 11 and 14 UT were averaged to match overpass time of TROPOMI above the stations. **TROPOMI data was filtered for the qa>0.5 (see Sub-section 2.1.2) and a radius of 5 km around SAOZ stations.** Figure 4 shows the evolution of the monthly mean and two standard error ($2\sigma$) of the tropospheric $NO_2$ columns above Paris and Guyancourt stations since January 2019 observed by SAOZ and TROPOMI (left panels). **The standard error corresponds to the standard deviation of the mean divided by the root number of considered days.***

*Figure 5: In some years, SAOZ observations in Guyancourt are higher, in some lower and sometimes they are very similar to those in Paris. Please discuss.*

For similar air masses similar results are expected. Effectively the years 2011, 2014, 2016 and 2018 show higher values at Guyancourt but the differences are considered as insignificant as remaining within the error bars (corresponding to the half of the 68% interpercentile of the median. The years 2011, 2014 and 2016 present slightly higher values between 0.1 and 0.8 Pmolec cm$^{-2}$. and 1.8 Pmolec cm$^{-2}$ in 2018. The higher values at Guyancourt are associated to days during weekend or holidays in more than 67% of the cases.

*Figure 5: Why are uncertainties in 2020 so much smaller than in other years?*

The error bars correspond to the IP68 of the median and in 2020, the values were low and much similar during the lockdown period and it was well sampled for our study since the wind direction and speed were favorable for our classification (see Figure 3 of the paper)

*Line 273: What is meant by "reweighted least squares with the bi-square weighting function)"?*

The idea using the reweighted bisquare function was to reduce the weight of the outliers' data far from the median fit calculated in the first place by a least square fitting.

The Line 273 was changed as follows:

*"… (reweighted bisquare function to reduce weight of outliers far ~5times from the median) …*

*Line 281: funding => finding*

Done

*Table 2: Non => No*

Done

*Line 321: I do not understand the reasoning about the lack of O3 for conversion of NO to NO2. While this may be the case close to large emission sources, I am not aware of a downward trend in O3, which could explain a difference in trends. In addition, as NOx emissions have reduced quite a lot over the last decade, this effect should be smaller now than 10 years ago. The similarity in trends at different altitudes of the Eiffel Tower is also not supporting the idea of slow conversion of NO to NO2.*

We thank the reviewer for this interesting remark.

Figure 6 shows that the three Paris urban background sites display $NO_2$ concentrations between 20 and 60 µg/m3. Figure 38 of the Airparif annual air quality report 2019 (Airparif et al. 2019), displayed here in Figure 6, shows three year average ozone levels at three urban background sites (left panel) varying from 35 to 43 µg/m$^3$ since 2005 and NO2 levels (right panel in Fig. 6 corresponding to Fig. 29 in AIPARIF report) varying from 41 to 33 µg/m$^3$ for background stations during the same period. Ozone and $NO_2$ levels of the same order of magnitude suggest incomplete NO to $NO_2$ conversion.

[Figure]

Figure 6. Evolution of the mean concentration of ozone of three background urban stations of Parisian agglomeration for 1992-1994 to 2017-2019 (Figure 38 of Airparif (2019)) and of mean concentration of $NO_2$ of six background urban stations (light blue) and five traffic stations (dark blue) for 1996-1998 to 2017-2019 (Figure 29 of Airparif (2019))

In such a situation, the $NO_2$ trends are impacted both by the NOx emission and ozone trends. Figure 38 cited above shows indeed strongly increasing ozone average urban background over Paris, for instance from 31 to 35 to 43 µg m$^{-3}$ respectively for the 1997-1999; 2007-2009 and 2017-2019 periods. This increase is the well-known counterpart of the NOx emission reductions (Airparif, 2019), but could also be due to global tropospheric ozone increases. We agree that the effect was even more pronounced some decades ago, but our data show that it is still present. We also agree with the referee, that our argument fails in explaining differences in trends between different in situ measurements. Given this we make the discussion a bit more detailed and equilibrated:

L321 to L324 were modified as follows:

*These trends appear to be less negative than those obtained from column measurements. Possible reasons for this are an increase of the $NO_2$ to NOx emission ratio, and a limitation by the available amount of $O_3$ for the NO to $NO_2$ conversion. Both factors affect more strongly the surface concentration than the boundary layer column, which could lead then to the different trend estimates.*

*Incomplete NO to $NO_2$ conversion is for example suggested by $NO_2$ and ozone concentrations of the same order of magnitude at Paris urban background sites (Figure 38 of Airparif 2019). In such a situation, the $NO_2$ trends are both impacted by the NOx emission and ozone trends. Figure 38 in Airparif (2019) cited above shows indeed strongly increasing ozone average urban background over Paris, for instance 35 to 43 µg m$^{-3}$ respectively for the 2007-2009 and 2017-2019 periods. This positive ozone trend buffers to some extent the negative NOx emission trend.*

*However while this reasoning would qualitatively explain differences in trends between column and in situ measurements, it fails to explain differences in trends between different in-situ sites, in the sense that larger NOx values would lead to smaller negative trends. This is not observed, on the contrary, the $NO_2$ trend is more negative at ground of Eiffel tower than at altitude when NOx becomes lower. Thus the exact explanation of differences in trends at different sites and heights still need more investigations*

Reference:
Airparif - Surveillance et information sur la qualité de l'air en Île-de-France – Bilan de l'année 2019, Juin 2020 (in French). Obtained in August 2021 on
https://www.airparif.asso.fr/sites/default/files/documents/2020-06/bilan-2019_0.pdf